# Interplay between the Chd4/NuRD Complex and the Transcription Factor Znf219 Controls Cardiac Cell Identity

**DOI:** 10.3390/ijms23179565

**Published:** 2022-08-24

**Authors:** Fadoua El Abdellaoui-Soussi, Paula S. Yunes-Leites, Dolores López-Maderuelo, Fernando García-Marqués, Jesús Vázquez, Juan Miguel Redondo, Pablo Gómez-del Arco

**Affiliations:** 1Institute for Rare Diseases Research, Instituto de Salud Carlos III (ISCIII), 28222 Madrid, Spain; 2Gene Regulation in Cardiovascular Remodelling and Inflammation Laboratory, Centro Nacional de Investigaciones Cardiovasculares Carlos III (CNIC), 28029 Madrid, Spain; 3Centro de Investigación Biomédica en Red de Enfermedades Cardiovasculares (CIBERCV), 28029 Madrid, Spain; 4Cardiovascular Proteomics Laboratory, Centro Nacional de Investigaciones Cardiovasculares Carlos III (CNIC), 28029 Madrid, Spain

**Keywords:** Chd4, NuRD, Znf219, chromatin remodeling, epigenetics, heart, arrhythmia

## Abstract

The sarcomere regulates striated muscle contraction. This structure is composed of several myofibril proteins, isoforms of which are encoded by genes specific to either the heart or skeletal muscle. The chromatin remodeler complex Chd4/NuRD regulates the transcriptional expression of these specific sarcomeric programs by repressing genes of the skeletal muscle sarcomere in the heart. Aberrant expression of skeletal muscle genes induced by the loss of Chd4 in the heart leads to sudden death due to defects in cardiomyocyte contraction that progress to arrhythmia and fibrosis. Identifying the transcription factors (TFs) that recruit Chd4/NuRD to repress skeletal muscle genes in the myocardium will provide important information for understanding numerous cardiac pathologies and, ultimately, pinpointing new therapeutic targets for arrhythmias and cardiomyopathies. Here, we sought to find Chd4 interactors and their function in cardiac homeostasis. We therefore describe a physical interaction between Chd4 and the TF Znf219 in cardiac tissue. Znf219 represses the skeletal-muscle sarcomeric program in cardiomyocytes in vitro and in vivo, similarly to Chd4. Aberrant expression of skeletal-muscle sarcomere proteins in mouse hearts with knocked down *Znf219* translates into arrhythmias, accompanied by an increase in PR interval. These data strongly suggest that the physical and genetic interaction of Znf219 and Chd4 in the mammalian heart regulates cardiomyocyte identity and myocardial contraction.

## 1. Introduction

Cardiovascular diseases (CVDs) are the most common cause of morbidity and mortality worldwide. Given the economic burden these conditions place on healthcare systems, research on this topic is therefore extremely important [1,2]. A prominent class of rare CVDs are the arrhythmias [3], mostly affecting the function of the cardiac conduction system (CCS). These disorders are very diverse phenotypically and their lethality depends on their origin and the level of involvement of the natural pacemaker and the CCS. The most sustained form of cardiac arrhythmia is atrial fibrillation (AF), which has a prevalence of 3% in the general population and is associated with cardiovascular morbidity, stroke, and ultimately sudden death [4,5].

The heart is one of the first organs to differentiate and function during mammalian development, since it must provide the embryo with oxygen from early stages. From its beginnings as a simple tube, the heart beats through synchronized cardiomyocyte contractions. In contrast, in the adult heart, the heartbeat is controlled by the CCS, which regulates and synchronizes the propagation of the electrical impulse from the atria to the ventricles. The electrical impulse arises in the sinoatrial node (SAN) in the right atrium, triggering the depolarization of neighboring cardiomyocytes and rapidly reaching the atrioventricular node (AVN), which results in the contraction of the atria (atrial systole). The AVN slightly delays the propagation of the electrical impulse, permitting late ventricular filling and the final steps of atrial contraction. From the AVN, the impulse travels through the asymmetrical bundles of His to the ventricles through the Purkinje fiber network that, in turn, will coordinate ventricular contraction and the ejection of blood through the right and left ventricular tracts [6,7]. This complex system is tightly regulated, and the slightest alteration to the synchronized propagation of electrical coupling could trigger arrhythmogenesis. Although some arrhythmias lead to sudden death, the mechanisms regulating their development remain poorly characterized.

The cardiac and skeletal muscles are both striated tissues that, despite their distinct embryonic origins, have a very similar contractile apparatus called the sarcomere. This highly organized structure is responsible for contraction and relaxation movements, and its cytoskeletal components are exquisitely assembled to generate contraction forces [8]. Despite the similarities between the sarcomeres of cardiac and skeletal muscles, the protein isoforms expressed in each tissue are different [9]. In each striated muscle type, transcriptional regulation of these isoforms is important for ensuring correct function and homeostasis. Abnormal expression of any sarcomeric protein can lead to a plethora of skeletal and cardiac myopathies. The regulation of striated muscles therefore requires the fine-tuning of numerous transcription factors and epigenetic regulators. These molecular networks maintain the correct expression of the specific isoforms required in any particular muscle and also repress the expression of the opposing transcriptional sarcomeric program. One of the epigenetic regulators involved in the maintenance of striated-sarcomeric identity is the chromatin remodeler Chd4 (chromatin helicase DNA binding protein 4, also called Mi-2β), which forms part of the chromatin remodeling complex NuRD (Nucleosome Remodeling and Deacetylation). Chd4 is a highly conserved ATP-dependent helicase involved in a wide range of cellular processes, such as transcriptional regulation, DNA-damage responses, cell-cycle progression, and eukaryote differentiation and development [10]. We and others recently reported that Chd4/NuRD regulates the identity of both striated muscle types by repressing the skeletal muscle program in the heart and the cardiac program in skeletal muscle [9,11]. The importance of this chromatin remodeler has also been revealed by the description of the human CHD4-related condition Sifrim–Hitz–Weiss syndrome (SIHIWES) [12,13,14]. This clinical research effort has revealed that mutations in CHD4 lead to congenital heart defects and neurodevelopmental disorders [15], strongly suggesting critical roles for this protein both in cardiac development and in neural development and plasticity.

The Chd4/NuRD complex does not directly bind to DNA, and research in other tissues has demonstrated that the complex is recruited to DNA by transcription factors [16,17,18]. In search of TFs that interact with Chd4 and recruit it to regulatory regions in cardiomyocyte DNA, we conducted an in silico analysis of putative TF DNA binding sites in targets ectopically expressed in *Chd4*-mutant hearts, using the TRANSFAC platform. This analysis showed that the transcripts upregulated in *Chd4*-mutant hearts were enriched in DNA-binding sites for the TF Znf219, among others. Znf219 (also called Zfp219) is a zinc-finger protein composed of nine Cys2His2-type zinc-finger domains [19,20]. The function of Znf219 is largely unknown, but it has been reported to act mainly as a repressor of transcription, like Chd4/NuRD [21]. Nevertheless, Znf219 can also positively regulate transcription, and enhances the activity of Sox9, which in turn regulates *Col2a1* transcription and chondrogenesis [22]. Moreover, the zebrafish Znf219 ortholog ZNF219L coordinates with Sox9a to regulate notochord-specific expression of Synuclein-2γ [23] and Collagen Type 2 Alpha 1a (Col2a1a) [24]. Dysregulation of Synuclein-2γ leads to accumulation of afunctional proteins that, in humans, lead ultimately to cellular dysfunction and neurotoxicity [23]. Znf219 has been shown to interact with NuRD complex components in several cell lines in vitro, and this interaction seems to involve the 4 N-terminal zinc-fingers of Znf219 and the C-terminal end of Chd4 [25].

The major aim of this study was to determine whether Znf219 and Chd4 interact in the heart, and whether this interaction regulates cardiac homeostasis. We show here that Znf219 and the NuRD complex also physically interact in the mammalian heart, strongly suggesting a genetic interplay between these proteins. These physical and genetic interactions led us to hypothesize that Znf219 might regulate, together with Chd4/NuRD, the repression of the skeletal-muscle gene program in the heart to ensure cardiomyocyte identity. Knockdown of *Znf219* in the HL-1 cardiomyocytic cell line and in mouse cardiomyocytes induced the ectopic expression of mRNAs and proteins of the skeletal muscle sarcomeric program, inducing heart arrhythmias.

## 2. Results

### 2.1. Promoter Regions of the Genes Upregulated in Chd4-Mutant Hearts Are Enriched in Znf219 DNA-Binding Sites

To decipher the transcriptional mechanisms controlling the repression of the skeletal muscle program in cardiomyocytes, we first conducted a bioinformatics analysis using our previously generated RNA-Seq data [9]. To identify TFs potentially regulating transcriptional repression, we screened the regulatory regions of the genes upregulated in adult PD21 Chd4^F/F^; Corin^cre/cre^ (Chd4^corin^) and embryonic (ED10.5) Chd4^F/F^; Nkx2-5^cre/wt^ (Chd4^nkx^) mutant hearts. An enrichment analysis using the TRANSFAC tool identified TFs potentially regulating the repression of these genes in wild-type (WT-Chd4^F/F^; Corin^wt/wt^ and Chd4^F/F^; Nkx2-5^wt/wt^) hearts (Figure 1). The promoters of these genes included TF DNA-binding sites (DBS) for cKROX, SP1/SP3, and Znf219, among others (Appendix A). We focused our study on Znf219 because it is the only one of these TFs that physically interacts with Chd4 ([25]; see the next section). Moreover, many of the upregulated genes in both the adult and embryo mutants were enriched in the Znf219 DBS (Figure 1a,b). The embryonic and adult mutant hearts showed similar and significant gene expression alterations, not only for the upregulated genes, but also for genes that were downregulated (Appendix A). These data suggest that Znf219 might regulate the repression of the skeletal-muscle sarcomeric gene program in the heart.

### 2.2. The Chd4/NuRD Complex and Znf219 Physically Interact in the Heart

The NuRD complex and Znf219 have been shown to pull down together in an in vitro system [25], and we confirmed this finding in co-transfected HEK293T cells (Appendix A). However, whether Chd4 and Znf219 interact directly in cardiac tissue has not been reported. We therefore first searched for proteins interacting with the NuRD complex in the adult heart by comparing the profiles of proteins co-immunoprecipitated with anti-Chd4 (α-Chd4) antibody in hearts from 4-week-old WT (Chd4^F/F^; Corin^wt/wt^) and Chd4^corin^ (Chd4^F/F^; Corin^cre/cre^) cKO mice. Mass spectrometry of Chd4-interacting proteins detected in WT tissue confirmed that many of the peptides were derived from NuRD complex components (Figure 1c and Appendix A). Anti-Chd4 immunoprecipitates from WT hearts also contained peptides derived from Znf219, which were not observed in Chd4^corin^ hearts (Figure 1c,d and Appendix A). This association was confirmed using Western blot (Figure 1e), demonstrating that Znf219 specifically interacts with the Chd4/NuRD complex in cardiomyocytes. Although anti-Chd4 immunoprecipitates included peptides derived from other TFs (eg, Zbtb20), no other TFs from the in silico study (such as SP1) were found to associate with Chd4 in the heart (Appendix A), suggesting that the physical interaction between Znf219 and Chd4 likely has transcriptional consequences.

### 2.3. Znf219 Is Highly Expressed in Mouse Heart and in Striated Muscle Cell Lines and Is Involved in the Repression of the Skeletal-Muscle Gene Program In Vitro

Considering that Znf219 interacts with Chd4 in the heart, we next analyzed the expression of this TF in cardiac tissue and other adult mouse organs, both at the mRNA ([22], Figure 2a) and protein levels (Figure 2b). The Znf219 transcript was highly expressed in the thymus and testis, followed by the liver, heart, and spleen (Figure 2a). Western blot analysis of selected organs showed the highest Znf219 protein expression in the heart (Figure 2b). The analysis of the transcriptional expression of Znf219 throughout heart development revealed that, while low at embryonic days 10.5 (ED10.5) and ED11.5, expression strongly increased at ED12.5. These levels were sustained during the following developmental stages until mouse adulthood (Figure 2c). We also tested the protein expression levels of Znf219 in the established skeletal-muscle cell line C2C12, where Znf219 expression is maintained at similar levels under growing and differentiation conditions (Figure 2d). Abundant Znf219 protein expression was also detected in the atrial cardiomyocytic cell line HL-1 (Figure 2d).

We next used an shRNA-lentiviral system to silence Znf219 or Chd4 in HL-1 cells and in this way analyzed whether the skeletal-muscle gene program is similarly repressed by each of these proteins (shRNA sequences in Appendix A). We first confirmed the cardiomyocytic-like identify of HL-1 cells using immunofluorescence detection of the cardiac markers alpha-actinin and cardiac tropomyosin. Up to 84% of cells expressed these cardiac markers (Appendix A), and the cultures beat coordinately when confluent (data not shown). Knockdown of Znf219 (Figure 2e, top) or Chd4 (Figure 2e, bottom) induced a statistically significant increase in the transcript levels of selected skeletal-muscle genes (Tnni2, Tnnc2, Atp2a1, and Tnnt3), with Chd4 knockdown having the stronger effect. Interestingly, Znf219 silencing reduced the expression of Acta1, in contrast to the effect of Chd4 knockdown or Chd4 cKO [9]. This different outcome could reflect differing levels of down-modulation with the lentiviral shRNAs, and we therefore checked Znf219 and Chd4 protein expression in HL-1 cells after silencing. This analysis showed that shRNA-Znf219 did not fully suppress Znf219 protein expression, contrasting the total lack of Chd4 protein after Chd4 knockdown (Figure 2f). Knockdown of Znf219 did not decrease Chd4 protein expression (Figure 2f), and Chd4 knockdown did not reduce Znf219 mRNA levels (Figure 2e and Appendix A).

These results confirmed that Znf219 and Chd4 both repress the expression of skeletal-muscle genes in cultured cardiomyocytes, also suggesting a putative genetic interaction between these proteins to negatively regulate skeletal-muscle gene promoters in the myocardium, whose ectopic expression would otherwise have deleterious consequences in vivo.

### 2.4. Znf219 Represses the Skeletal Muscle Gene Program in the Heart

The lack of Znf219 KO mice prompted us to knockdown this TF in vivo using shRNAs expressed by adeno-associated virus (AAV). We used the AAV serotype 9 (AAV9), which efficiently and stably transduces the heart [26]. On day 1 or 2 after birth (PD1 and PD2), mouse pups were given subcutaneous injections of shRNA-Znf219-eGFP-U6 (shRNA-Znf219—check sequences in Appendix A) or AAV-shRNA-Scramble-eGFP-U6 (shRNA-Scramble) at a dose of 2 × 10^11^ viral particles per pup. To test which shuttle shRNA-Znf219 plasmid worked best in vivo, we first tested the three commercially available plasmids (Appendix A). While all three Znf219-targeting shRNAs had a similar silencing effect at the RNA level (Appendix A), shRNA1 had the best transduction efficiency, as shown by immunohistochemistry (IHC) detection of GFP (Appendix A), and we therefore used this shRNA for further analysis.

The infected mice were monitored using electrocardiography between the ages of 4 and 8 weeks, followed by euthanasia and sample collection for further study (Figure 3a). Fluorescence microscopy of dissected hearts confirmed extensive GFP expression, both in the atria and to a lesser extent in the ventricles (Figure 3b). Closer inspection of heart morphology revealed a slight increase in size of hearts infected with shRNA-Znf219. Determination of the heart weight/tibia length ratio confirmed that sh-Znf219-infected hearts were hypertrophic when compared to those of sh-Scramble-infected mice (Figure 3c). IHC detection of GFP expression demonstrated extensive yet mosaic infection throughout the heart chambers (Figure 3d). H&E staining revealed no apparent morphological changes in the hearts of sh-Znf219-infected mice (Figure 3d); however, in 30% of the animals (4 out of 13) knockdown for Znf219 presented apparent interstitial fibrosis (Figure 3d and Appendix A). To further analyze fibrosis, we quantified collagen deposition with MetaMorph offline software. In these experiments, we observed no significant increase in collagen deposition; however, an increase tendency of collagen could be observed (Appendix A). In order to deepen in the collagen deposition in ShZnf219 knocked down hearts versus shScramble, we next tested the levels of expression of the transcripts for Collagen type I alpha 1 (Col1a1) and collagen type III alpha 1 (Col3a1) and found a tendency of increase in Col1a1 (*p* = 0.08) and a significant increase in Col3a1 (Appendix A).

Znf219 RNA knockdown and protein reduction were confirmed using RT-qPCR (Figure 4a) and Western blotting (Figure 4b), respectively. To confirm the in vitro data in HL-1 cells (Figure 2), we next checked the transcriptional expression state of sh-Znf219 infected and sh-Scramble-infected hearts. This analysis showed that Znf219 knockdown induced ectopic expression of some of the same skeletal-muscle sarcomeric transcripts upregulated in Chd4 cKO mice [9]: Acta1 (2.1-fold), Tnnc2 (2.5-fold), and Tnnt3 (2.5-fold) (Figure 4c). Atp2a1 and Tnni2 did not appear to be upregulated, but this likely reflects the lower detection background, since the AAV9 infection was mosaic. We also analyzed whether RNA upregulation in vivo translated into protein expression. In contrast to its corresponding mRNA, we could detect ectopic protein expression of troponin I2, fast skeletal type (Tnni2) in Znf219-knockdown hearts, as well as the troponin I3, fast skeletal type (Tnnt3) isoform (Figure 4d). Like Tnni2, Atp2a1 mRNA was not apparently upregulated, but we nonetheless detected scattered Atp2a1 protein expression in atrial cardiomyocytes, likely reflecting mosaic infection (Figure 4e). Given the involvement of Chd4 in regulating CCS genes and calcium, sodium, and potassium ion channels [9], we investigated whether Znf219 knockdown produced similar outcomes, but we observed no changes relative to sh-Scramble hearts (Appendix A).

Finally, given that Znf219 represses the expression of the skeletal-muscle genes in the heart, we wanted to discard a concomitant transcriptional alteration of sarcomeric cardiac genes. qPCRS analysis of some cardiac muscle genes, indeed, discarded this hypothesis (Appendix A).

### 2.5. Cardiac Znf219 Knockdown Induces Arrhythmia in Adult Mice

Chd4 cardiac deletion induces arrhythmias, probably due to contraction defects provoked by the misexpression of skeletal-muscle sarcomeric isoforms [9]. For this reason, we next checked whether the skeletal muscle gene misexpression in Znf219 KD hearts also presented alterations in their electrocardiograms (Figure 5), as it does in Chd4 cKO mice [9]. While the ECGs of mice transduced with ShScramble were normal 4 weeks after infection (Figure 5a), 68% of mice transduced with shRNA-Znf219 developed arrhythmias (Figure 5a). At this age, all animals had normal atrial (*p* wave) and ventricular (QRS) depolarization (Figure 5a). By 8 weeks of age, while ShScramble-infected mice had normal ECGs (Figure 5b), up to 78% of Znf219-knockdown mice had more prominent arrhythmias (Figure 5b and Appendix A), with episodes of bradycardia and tachycardia, suggesting a progressive phenotype. Again, QRS waves were normal (Figure 5c, right), but the PR interval spiked in the Znf219-knockdown mice at this age (Figure 5c, left). This finding indicates a deceleration of conduction between the atria and ventricles, which is usually due to slow conduction through the AVN, thus, reflecting first-degree atrioventricular block.

## 3. Discussion

The results presented here show that Znf219 physically interacts with Chd4/NuRD in the heart and demonstrates a putative cooperation between these proteins in the control of cardiomyocyte cell identity, in part through the repression of the skeletal-muscle gene program. We previously reported that Chd4/NuRD regulates striated muscle identity by repressing the opposing sarcomeric program. Thus, Chd4/NuRD represses the skeletal-muscle sarcomeric program in the heart, as well as regulating the metabolic performance of this tissue [9]. In search for TFs interacting with and potentially recruiting the chromatin remodeling complex NuRD to skeletal-muscle gene promoters in the heart, we first performed a TRANSFAC in silico study to analyze the promoter regions of the genes found previously to be upregulated in Chd4 cKO hearts [9]. This analysis showed that many of these gene promoters contained putative DNA-binding sites for Znf219, together with DNA-binding sites for other TFs, such as SP1 and KROX.

Supporting a role for Znf219 in the cardiovascular system, we observed high expression of Znf219 transcripts and protein in the heart in comparison with other tissues (Figure 2, HumanAtlas (https://www.proteinatlas.org/ENSG00000165804-ZNF219/tissue, accessed on 20 August 2022)). We also observed that *Znf219* mRNA expression increases sharply in embryonic hearts from ED11.5 to ED12.5, and that this elevated expression is then maintained at later stages of development and after birth. The elevation of *Znf219* mRNA levels during the later stages of development suggests that this repressive TF plays a prominent role in the heart once this organ is formed, but that it may not be necessary for early stages of heart development. Supporting this view, *Znf219* knockdown in the adult heart induced the ectopic expression of some skeletal-muscle sarcomeric genes, which resulted in the induction of arrhythmias similar to those observed in *Chd4* cKO mice [9]. In our study, transduction with the *Znf219* AAV9-shRNAs was efficient but always mosaic, which would explain a milder phenotype. Likely as a consequence of this, we did not detect upregulation of *Tnni2* or *Atp2a1* mRNA in hearts infected with AAV9-shRNA-*Znf219*, contrary to our findings in HL-1 cells silenced with lentiviral shRNA-*Znf219*, even though the *Znf219*-knockdown hearts showed clear evidence of ectopic expression of both Tnni2 and Atp2a1 proteins; ectopically expressed Tnni2 was detected in *Znf219*-knockdown hearts with Western blot, and Serca1 (Atp2a1) was prominently expressed in atria, with a patchy pattern, reflecting the mosaic viral transduction. In contrast, *Chd4* lentiviral silencing in the cardiomyocytic cell line HL-1 induced a strong upregulation of some of the transcripts examined, including *Tnnc2* (more than 30 times) and *TnnT3* and *Atp2a1* (more than 20 times) compared with *Znf219*-silenced cells. This finding corroborates our previous report in *Chd4* cKO mice [9]. Notably, Western blotting analysis showed complete knockdown of Chd4 but only a partial reduction in Znf219 protein, which may explain the partial phenotype of Znf219-silenced HL-1 cells. This reflects a direct transcriptional effect of Znf219 on the upregulated genes, since no downregulation of *Chd4* mRNA or Chd4 protein was observed in Znf219-silenced HL-1 cells. The lack of *Znf219* cKO models and the potential for compensatory action by other zinc-finger proteins make it difficult to study the full contribution of this TF to heart homeostasis and heart development. Nevertheless, our approach highlights the importance of Znf219 in the heart and opens the way to new research into the function of this TF in heart homeostasis.

It was recently reported that the transcription factors Gata4, Nkx2-5, and Tbx5 physically interact with Chd4, recruiting it to repress the expression of tissue-specific genes during cardiac development [27]. Our proteomic analysis of Chd4 immunoprecipitates from adult heart tissue did not detect peptides of any of these heart-differentiation TFs, suggesting that Znf219 may be necessary for the repression of these genes in the fully differentiated tissue. As mentioned before, the upregulation of *Znf219* during heart development might corroborate these data. One can envision a scenario in which the developmental TFs Gata4 and Nkx2-5 recruit the NuRD complex during development, but that Znf219 takes over this role at later stages of development and in the adult, perhaps with additional involvement of other ubiquitously expressed TFs, such as SP1, MAZ, or WT1. In fact, we found SP1 and MAZ DNA-binding sites in the promoters of many of the genes upregulated upon Chd4 cKO. These G-rich DNA-binding sites are interchangeable, and so can be bound by any of these TFs, including Znf219 [28]. Despite the difficulties posed by the lack of Znf219 cKO mice, it will still be important to study the implication of Znf219 in these processes in order to ascertain whether it cooperates with cardiogenic TFs during development and whether Znf219 or other TFs maintain the repression of the skeletal-muscle sarcomeric gene program in the post-mitotic heart, and of the cardiac sarcomeric gene program in skeletal muscle.

To analyze the pathological implications of Znf219 knockdown in the heart, we conducted electrocardiographic examination of infected mice, finding that the PR interval was significantly increased in shRNA-*Znf219*-infected animals. This is indicative of a delay in the propagation of the electrical impulse from the atria to the ventricles, although ventricle contraction seemed normal. The relatively greater impairment of atrial function compared with the ventricles may indicate a higher rate of infection in the atria, where we detected ectopic expression of Atp2a1, in contrast with the ventricles where we did not. Atp2a1 plays a pivotal role in Ca^2+^ regulation in skeletal muscle, and Atp2a1 mutations have been linked to Brody myopathy and atrophic muscular disease [29]. Overexpression of this protein and of the skeletal-muscle troponins could lead to arrhythmias, but this was not sufficient to lead to heart failure in our experimental setting.

In addition to Znf219, our proteomic analysis also detected binding of other TFs, for example, Zbtb20 (Appendix A). Lack of Zbtb20 causes metabolic and cardiac contractile dysfunction [30], and this TF seems to regulate the expression of mitochondrial genes, which are also a target of Chd4/NuRD in the heart. Further research will be needed to determine whether the regulation of metabolic genes and potentially other targets is mediated by the interaction between Zbtb20 and Chd4. Regarding other transcriptional regulators that might counteract the actions of the Chd4/NuRD complex, we cannot discard myogenic transcription factors. For example, Mef2 and Srf are expressed in both striated muscles, where they positively regulate the transcription of sarcomeric proteins. Chd4/NuRD and Znf219 may contribute to maintenance of closed chromatin surrounding DNA-response elements for Mef2, Srf, and other TFs in skeletal-muscle genes in the heart, thus, preventing their binding. More research will be needed to fully demonstrate that Znf219 recruits Chd4/NuRD or maintains its binding to repressed genes in the heart and skeletal muscle, where Znf219 is also highly expressed. The lack of good Znf219 ChIP antibodies has so far prevented us from checking this possibility.

Altogether, our data indicate that Znf219 regulates cardiomyocyte cell identity, by controlling the transcriptional repression of the skeletal muscle gene program. Therefore, knockdown of this Chd4-interacting transcription factor results in arrhythmias, with a significant increase in the PR interval, a phenotype very similar to that observed in Chd4 cKO mice. In our working model (Figure 6), Znf219 and Chd4/NuRD, probably through their interaction, maintain the repression of skeletal muscle genes in a differentiated cardiomyocyte (Figure 6a). This would maintain the normal physiology, contractility, and energy intake of the cell. When Chd4 or Znf219 are removed from the cardiomyocyte (Figure 6b), the skeletal muscle sarcomeric program is aberrantly expressed in the cardiomyocyte and this translates to the induction of arrhythmias, cardiac fibrosis, and, eventually, to heart failure.

### Study Strengths and Limitations

The lack of *Znf219* cKO mice makes very difficult to study the role of this transcription factor in vivo. Our approach to the problem, infecting with adeno-associated viruses to knock down the expression of this gene, allowed us to conclude that Znf219 is important in heart homeostasis, keeping skeletal muscle genes silenced. The limitations of the study, however, can be found in the translation of our results to human cardiac pathophysiology, since our investigation is basic in nature. Chd4 mutations have been found in humans that translate into cardiac disorders, but more research would be needed in order to find mutations or misexpression of *Znf219* in humans. Another limitation of our study is that, although we found a direct interaction between Znf219 and Chd4, we did not directly demonstrate in this work that there is a transcriptional cooperation between both proteins. We are currently performing *Znf219* silencing experiments in *Chd4* cKO mice to corroborate this hypothesis.

Our study highlights the power of reanalyzing existing transcriptomics datasets using appropriate bioinformatic tools. For the discovery of novel factors, we envision that this approach will be boosted in the near future with the use of more advanced statistical tools optimized to extract relevant information depending on the characteristics of the transcriptomics datasets, such as artificial intelligence models.

## 4. Materials and Methods

### 4.1. Mouse Strains and Adeno-Associated Viral Infections

The Chd4 floxed and transgenic Corin cre/cre mouse strains have been described previously [9,18]. Wild-type (WT) adult male and female mice on the CD1 genetic background were originally obtained from Jackson Laboratories (Charles River Laboratories) and housed in the CNIC animal facility. Pups (P0–P1) were ice-anesthetized, immobilized, and given subcutaneous injections of 20 μL AAV9-shRNA-Scramble or AAV9-shRNA-*Znf219* viral particles (VP) at 2.1 × 10^11^ VP/mL. The subcutaneous injection was administered in the dorsal region of the pup with a micro-fine insulin syringe [26]. Correct injection was verified by bump formation beneath the skin. After the injection, pups were allowed to recover for several minutes on a heated cage at 37 °C before being returned to their mothers.

### 4.2. AAV9 Generation

AAV vectors were produced by the triple transfection method using HEK293T cells [31,32]. The shuttle plasmids pAAV[shRNA]-EGFP-U6 > mZnf219[shRNA#1] or [shRNA scramble] (VectorBuilder) were packaged into AAV9 capsids using the helper plasmids pAdDF6 and plasmid pAAV9 (providing the rep and cap viral genes) (PennVector). All plasmids were co-transfected into HEK293T cells using linear polyethylenimine (MW 25,000). The cells were seeded in Hyperflasks (Corning) at 1.2 × 10^8^ cells per flask the day before transfection.

Transfection and viral particle collection were performed as previously described [33]. Sequences of sh-RNAs are included in Appendix A.

### 4.3. Electrocardiography (ECG)

ECG was performed on 4–8-week-old sedated mice using Accutac Diaphoretic ECG Electrodes (ConMed Corp., Utica, NY, USA) [9].

### 4.4. Protein Identification Using Mass Spectrometry

Whole WT (Chd4^F/F^; Corin^wt/wt^) and Chd4 (Chd4^F/F^; Corin^cre/cre^) mutant hearts were dissected in PBS and nuclear extracts were obtained [9]. Chd4 was immunoprecipitated with a mouse monoclonal anti-Chd4 antibody (ab70469, Abcam, Cambridge, UK). Proteins were in-gel digested with trypsin at a 5:1 protein: trypsin (*w*/*w*) ratio. Peptides were analyzed using LC-MS/MS [34].

The MS/MS raw files were searched against the Mouse Swissprot database using Sequest running in Proteome Discoverer 1.4. Peptide identification was validated using the probability ratio method [35] with an additional filtering for precursor mass tolerance of 12 ppm [36]. The false discovery rate (FDR) was calculated using decoy databases. Peptide and scan-counting were performed assuming as positive events those with an FDR equal to or below 5%.

### 4.5. Cell Culture, Lentiviral Production, and Cell Infection

The HL-1 cell line was maintained in Claycomb medium (Sigma-Aldrich, Merck KGaA, Darmstadt, Germany) supplemented with 10% FCS, L-glutamine (2 mM), penicillin (100 U/mL), and streptomycin (100 U/mL) in standard culture conditions. Before use, complete Claycomb medium was supplemented with insulin, retinoic acid, ascorbic acid, and norepinephrine [37]. This supplemented medium was added to the cells every 24 h, and the cells were re-seeded every three to four days, once they had become confluent and started beating.

Non-differentiated cells of the C2C12 cell line were maintained in Growth Medium (GM—Dulbecco’s modified Eagle medium (DMEM, Sigma) supplemented with 20% fetal bovine serum (FBS)), and differentiated C2C12 cells were maintained in Differentiation Medium (DM—DMEM supplemented with 2% horse serum (Gibco, Thermo Scientific, Cambridge, UK) and insulin). GM was added to the cells every 24 h, and the cells were re-seeded every three to four days, once they had reached 30–40% confluency. DM was added to induce differentiation of cells to myotubes when cells were confluent.

Lentiviruses encoding short hairpin RNAs targeting *Chd4* and *Znf219* (SIGMA-Aldrich) were produced by transient calcium phosphate transfection of HEK-293 cells with a three-plasmid HIV-derived and VSV-pseudotyped lentiviral system (Addgene plasmid 12259). HEK-293 cells were cultured in DMEM (Sigma) supplemented with 10% fetal bovine serum (FBS), L-glutamine (2 mM), penicillin (100 U/mL), and streptomycin (100 U/mL). Cells were plated at 30% confluency and transfected the next day. At 48 and 72 h after transfection, supernatants were collected, concentrated by ultracentrifugation (88,000 g for 2 h at 4 °C), and stored at −80 °C. HL-1 cells were then infected with concentrated lentiviral supernatant for 24 h. Then, 24 h later, puromycin was added to select infected cells and RNA and protein were extracted 48 h later for analysis. Sequences of sh-RNAs are included in Appendix A.

### 4.6. Transient Transfection with Calcium Phosphate

The HEK293 cell line was used for these experiments because of its high transfection efficiency. Cells were prepared and cultured to 80% confluency. Cells were plated on the evening before the transfection day. The next day, at least 4 h in advance of transfection, the medium was changed. One plate was transfected with Flag-Chd4 plasmid DNA and two plates were transfected for co-transfection with Flag-Chd4 and Myc-Znf219 plasmid DNA.

Transfection mixtures were prepared for each 150 mm^2^ plate in 15 mL FALCON tubes. The transfection mixture for Chd4 transfection included 1.25 mL HBS (Hepes Buffered Saline) and 20 µg DNA. The mixture for co-transfection with Chd4 and Znf219 included 2.5mL HBS (Hepes-buffered saline), 60 µg Chd4 plasmid DNA, and 20 µg Znf219 plasmid DNA. CaCl_2_ (150 µL) was added dropwise to each mixture with gentle vortexing. This procedure favors the formation of precipitates during a 20 min incubation at room temperature, after which mixtures were uniformly added to the cells. The cells were placed in the incubator overnight at 37 °C, 5% CO_2_. The next day, the condition of the cells was checked, and transfection was stopped by removing the transfection medium and adding 20 mL of fresh medium.

### 4.7. Reverse Transcription-qPCR (RT-PCR) and Gene Expression Analysis

RNA from cell cultures was collected by adding 1 mL of TRIsure buffer (Ecogen) to the culture plate after washing with PBS. Embryonic hearts of CD1 WT mice were collected, individually introduced into 1 mL TRIsure buffer, immediately snap-frozen in liquid N_2_, and stored at −80 °C for later RNA extraction. Hearts from CD1 mice transduced with AAV9-shRNA-Scramble or AAV9-shRNA-*Znf219* (VectorBuilder) were collected in 1mL of Trisure for RNA extraction.

Tissues were homogenized with a Magnalyzer system, and total RNA was isolated according to the manufacturer’s instructions. Reverse transcription reactions were carried out with 2 μg total RNA and using standard procedures. Quantitative PCR was performed on a CFX96TM Real Time System and aC1000TM Thermal Cycler using SYBR^®^-Green and the gene-specific oligonucleotide primers (Table 1).

### 4.8. Western Blotting

For detection of Chd4, Znf219, troponin I2, fast skeletal type (Tnni2), and troponin I3, fast skeletal type (Tnnt3), heart tissue was homogenized in lysis buffer (50 mM Tris pH 8, 0.42 M NaCl, 1% Triton X-100, 1 mM EDTA) supplemented with protease inhibitors (pepstatin, leupeptin, aprotinin, and PMSF) using a Magnalyzer apparatus. The resulting homogenate was incubated for 60 min at 4 °C on a rocking platform and then centrifuged at 14,000× *g* for 15 min, and the supernatant was recovered. Equal amounts of protein were resolved by 6% or 10% SDS-PAGE, and proteins were transferred to 0.22 µm nitrocellulose membranes. Blots were probed with polyclonal primary antibodies against Chd4 (1:1000 dilution; ab72418, Abcam, Cambridge, UK), Znf219 (1:1000 dilution; ab71279, Abcam), Tnni2 (1:1000 dilution; ab119943, Abcam, Cambridge, UK), or Tnnt3 (1:1000 dilution; sc-20643; Millipore, Bedford, USA), followed by HRP-conjugated goat anti-rabbit IgG secondary antibody (Bio-Rad) and detection by enhanced chemiluminescence (Amersham, Los Angeles, CA, USA). As a loading control, membranes were re-probed with a primary antibody against tubulin (mouse monoclonal, 1:40,000 dilution; T6074, Sigma) or PSF (mouse monoclonal, 1:1000 dilution; Santa Cruz Biotechnology, Dallas, TX, USA) (Table 2).

### 4.9. Histological Analysis

Adult hearts were dissected in cold PBS, fixed overnight in 4% paraformaldehyde buffer, embedded in paraffin, and cut at 5 μm intervals with a microtome. Sections were stained with hematoxylin and eosin (H&E) using standard procedures. Immunohistochemistry (IHC) was performed using standard procedures.

All images were acquired at room temperature using a DM2500, Leica microscope fitted with 10, 20, or 40 × HCX PL Fluotar objective lenses and Leica acquisition software (Leica Application Suite V3.5.0). Images were processed for presentation with the Adobe Photoshop program.

### 4.10. Transcription Factor (TF) Binding Analysis

Putative TF binding motifs were detected in the regions 600 bp upstream and 200 bp downstream from the transcription start sites (TSS) of genes of interest with the STORM algorithm [38], using position frequency matrices (PFMs) from the TRANSFAC database (Professional version, release 2009.4) [39]. The cut off *p*-value was 0.0000125. To identify over-represented TF binding sites in the dysregulated genes against the background of all genes in the genome, an enrichment analysis and heat mapping were performed using Gitools [40]. Resulting *p*-values were adjusted for multiple testing using the Benjamin and Hochberg’s false discovery rate (FDR) method [41].

### 4.11. Statistical Analysis and Software Used

Results are expressed as mean ± S.E.M. The significance of differences between WT and corresponding mutant groups was assessed by two-tailed Student’s *t*-test or by one-way or two-way ANOVA, as stated in each figure legend. Statistical significance was set at a *p* value < 0.05.

The software used in this study was GraphPad Prism 8.0 for quantitative data analysis, Adobe Illustrator CC 2018 to generate the figures, and MetaMorph offline software (University Image Corp., Buckinghamshire, UK) for fibrosis quantification.

## Figures and Tables

**Figure 1 ijms-23-09565-f001:**
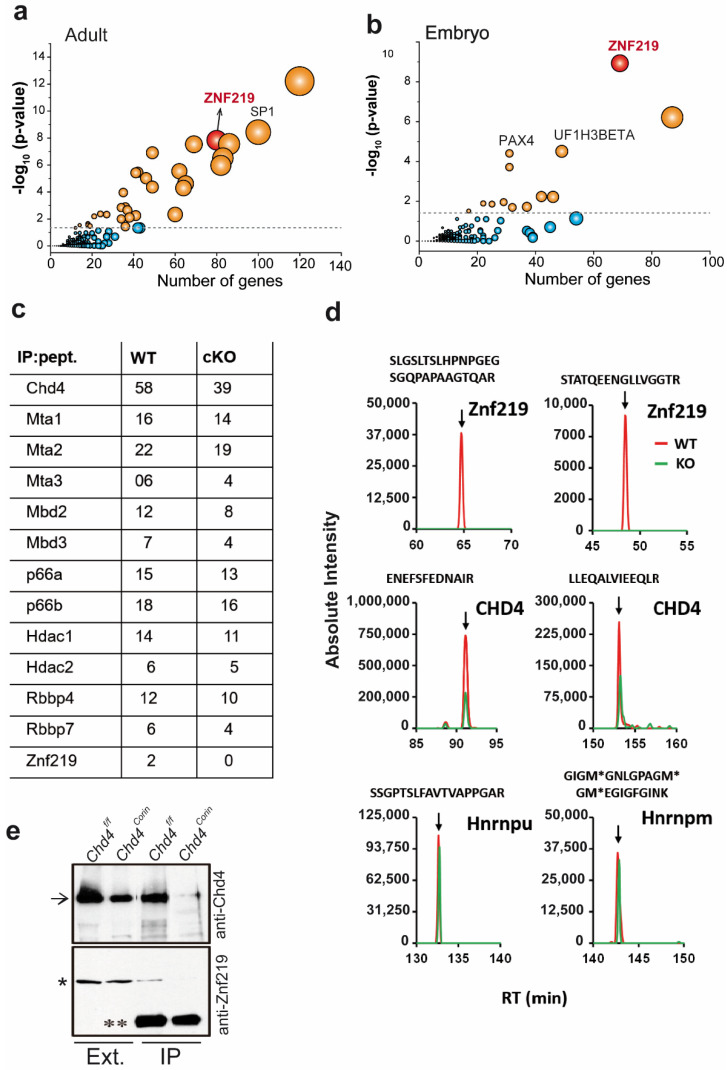
Chd4 and Znf219 physically interact in the mouse heart. (**a**) Bubble plot representation of the enrichment analysis for transcription factor (TF) binding sites in the set of upregulated genes in adult Chd4^corin^ hearts (TRANSFAC version 7.4). (**b**) Bubble plot representation of similar enrichment analysis for TF binding sites present in the set of upregulated genes in ED10.5 Chd4^nkx^ hearts. (**c**) The Chd4/NuRD complex composition in heart tissue; components were identified from peptides (pept.) recovered by mass spectrometry after Chd4 immunoprecipitation (IP) from 4-week-old WT mouse hearts (*Chd4^F/F^*; *Corin^wt/w^*). The table represents the number of distinct peptides (IP: pept.) derived from the proteins associated with Chd4 (Mi-2β) in WT hearts, and in *Chd4^corin^* (*Chd4^F/F^*; *Corin^cre/cre^*) cKO hearts. (**d**) The specificity of the association of Znf219 with WT Chd4 was demonstrated by analyzing the extracted ion chromatography traces of the peptides from the proteins indicated in each graph. M* means oxidated methionine. (**e**) Western blot of WT and *Chd4^corin^* cardiac nuclear extracts (Ext.) and Chd4 (α-Chd4) immunoprecipitated (IP), showing an association between Chd4 (arrow) and Znf219 (*)) only in WT hearts; ** = IgG heavy chain from denaturalized IP antibody. The double asterisk marks the heavy chain of antibodies used for IP.

**Figure 2 ijms-23-09565-f002:**
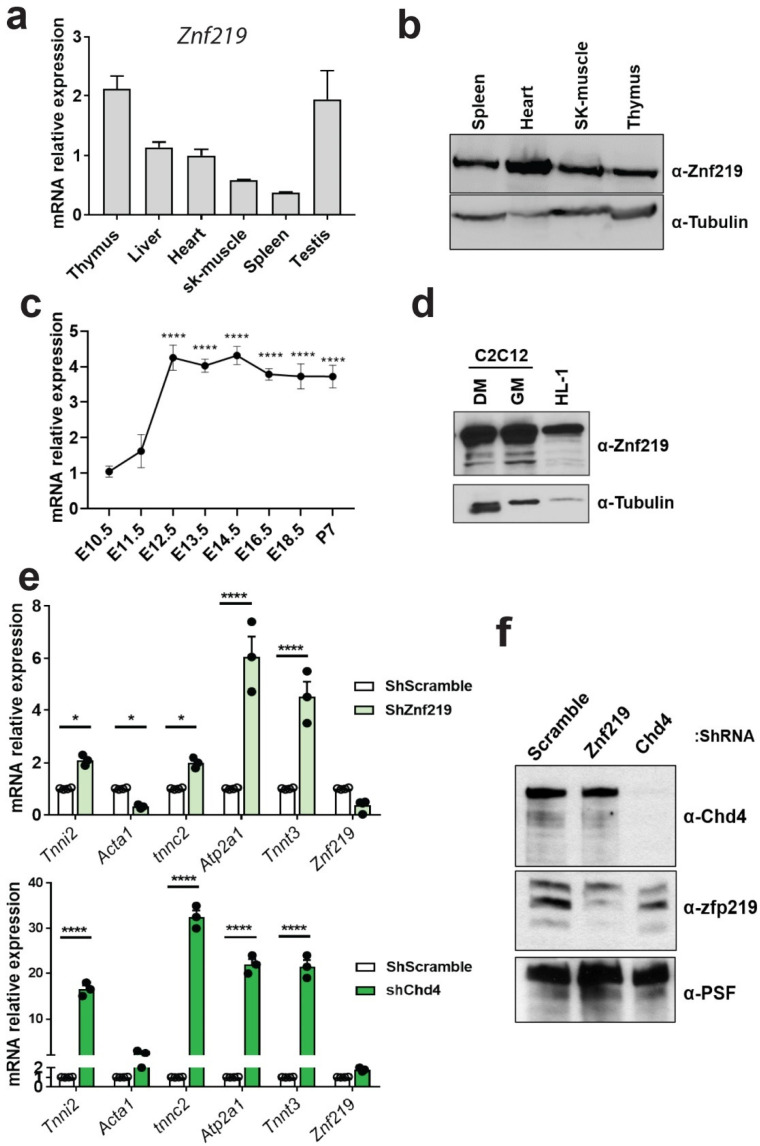
Znf219 represses the skeletal-muscle sarcomeric program in vitro. (**a**) Relative mRNA expression of Znf219 in different mouse organs (n = 4). Sk-muscle = skeletal muscle (tibialis anterior). (**b**) Western blot of different mouse organs, showing predominant Znf219 protein expression in the heart. (**c**) Relative mRNA expression of Znf219 during mouse cardiac development (n = 3). (**d**) Western blot of skeletal and cardiac cell lines, showing Znf219 protein expression in myocytic cell line C2C12 and the cardiomyocytic cell line HL-1. DM = differentiation media; GM = growth media. (**e**) qPCR analysis of representative skeletal muscle genes in HL-1 cells silenced for Znf219 or Chd4 with corresponding shRNAs. Cells infected with scrambled shRNA were used as a control (n = 4); two-way ANOVA (Bonferroni); * adj-*p*-value < 0.05, **** *p*-value < 0.0001. (**f**) Western blot showing the extent of knockdown of Znf219 (lane 2, middle) and Chd4 (lane 3, up) with the corresponding shRNAs. Lane 1 corresponds to HL-1 cells infected with scrambled shRNA (Scramble).

**Figure 3 ijms-23-09565-f003:**
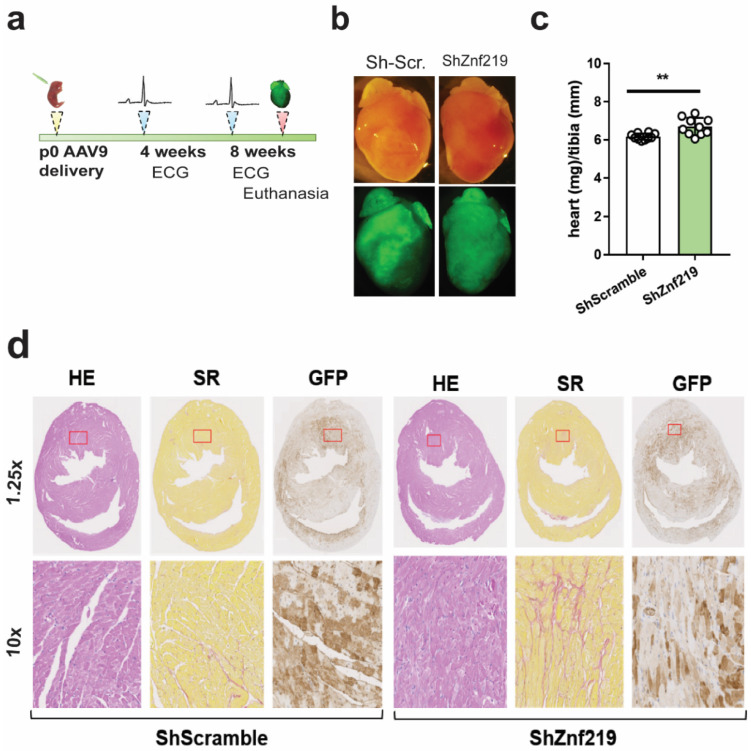
Znf219 regulates heart homeostasis in vivo. (**a**) Schematic representation of the experimental design after subcutaneous AAV9 injection into CD1 PD1-2 pups. (**b**) Gross morphological appearance of whole hearts of 8-week-old infected mice (upper panels) and fluorescence microscopy showing GFP expression in both atria and ventricles (lower panels). (**c**) Heart weight/tibia length ratios in infected mice. Paired *t*-test (two-tailed), ** adj. *p*-value < 0.01. (**d**) H&E (HE), Sirius Red (SR), and GFP staining on transverse heart sections from 8-week-old infected mice. Hearts were infected with 2 × 10^11^ particles of AAV9-shRNA-Scramble (left) or 2 × 10^11^ AAV9-shRNA-Znf219 (right). Mice infected with shRNA-Znf219 show interstitial fibrosis (middle right). ShRNA-Scramble (n = 9); shRNAZnf219 (n = 13).

**Figure 4 ijms-23-09565-f004:**
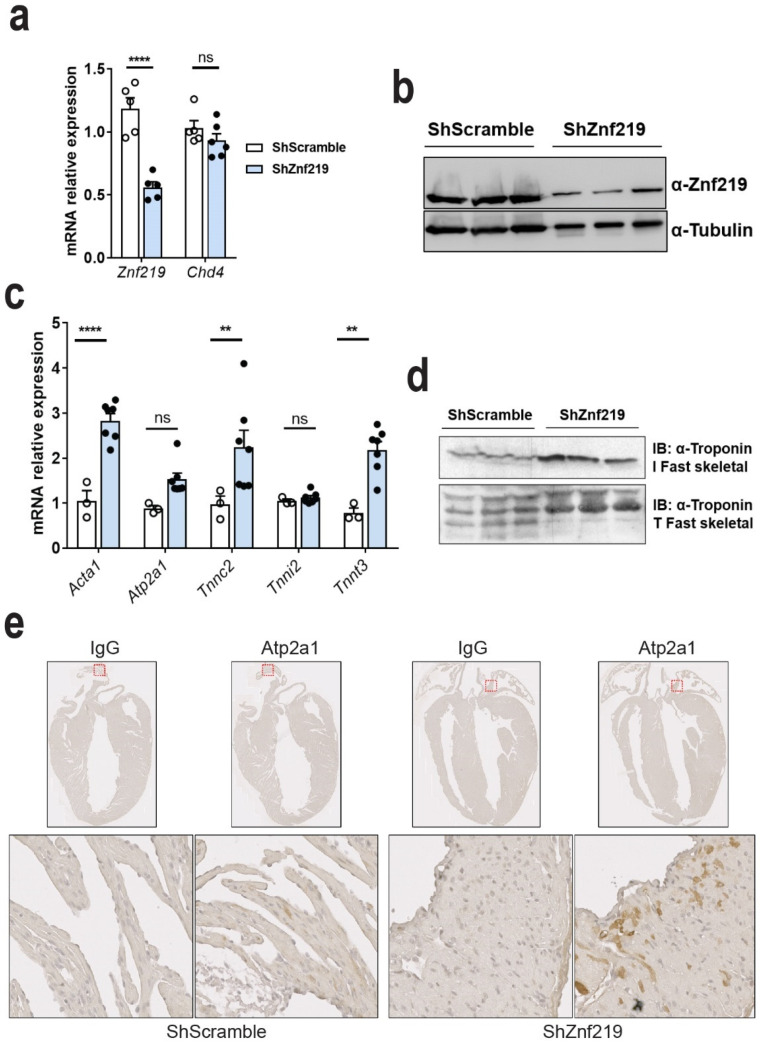
Znf219 represses the skeletal-muscle sarcomeric program in vivo. (**a**) qPCR analysis showing downregulation of Znf219 but no change in Chd4 in mice infected with ShZnf219; paired t-test (two-tailed) **** *p*-value < 0.0001). (**b**) Western blot showing Znf219 protein expression in ShScramble- and ShZnf219-infected mice. (**c**) qPCR analysis of selected genes encoding sarcomeric skeletal-muscle proteins (two-way ANOVA: ns = not significant; ** *p*-value ≤ 0.01, **** *p*-value ≤ 0.0001); ns = non-significant. (**d**) Western blot showing α-troponin I and T fast skeletal protein expression. (**e**) IHC analysis of Atp2a1 in longitudinal heart sections of 8-week-old infected mice. ShZnf219 mice (right) show ectopic atrial expression of Atp2a1.

**Figure 5 ijms-23-09565-f005:**
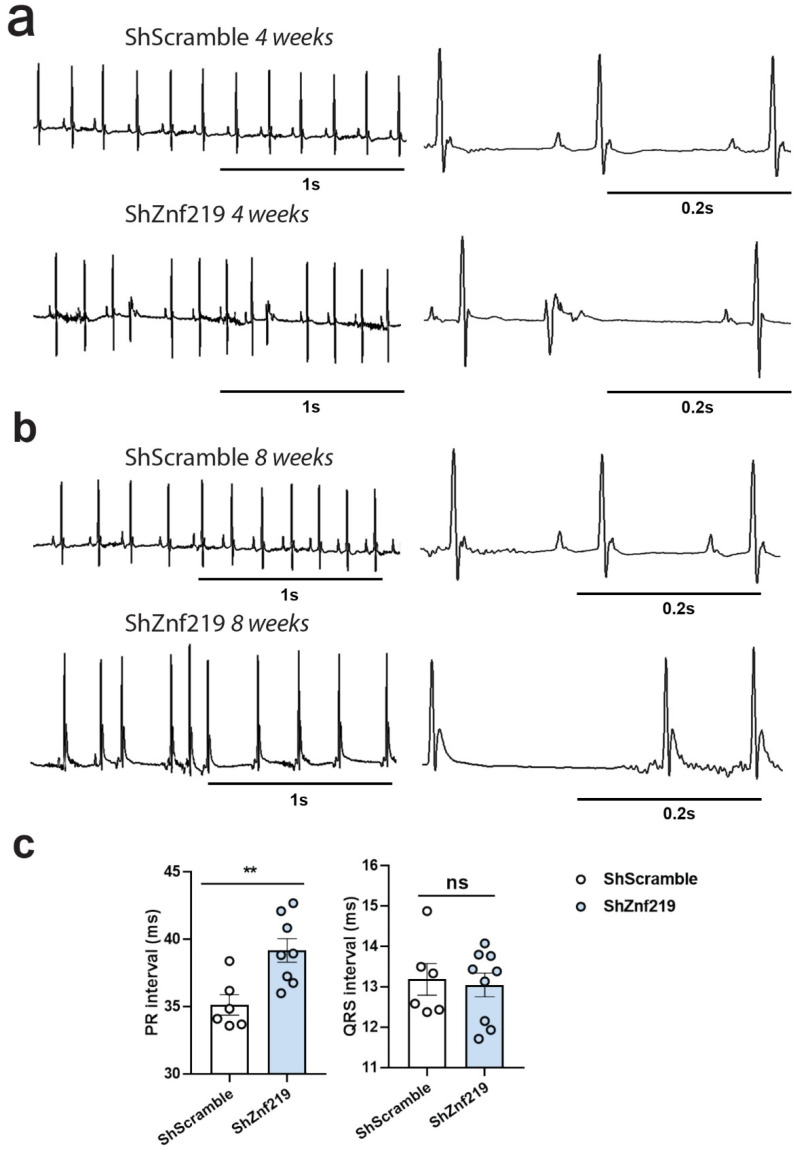
*Znf219* knockdown induces cardiac arrhythmias. (**a**) Electrocardiography (ECG) traces of 4-week-old and (**b**) 8-week-old mice infected with shRNA-Scramble or shRNA-*Znf219*, the latter showing arrhythmias (**c**). PR and QRS intervals in 4-week-old mice infected with shRNA-Scramble (*ShScramble*) or shRNA-*Znf219* (*ShZnf219*)., ** *p* ≤ 0.01, (mean + S.E.M.), ns = non-significant.

**Figure 6 ijms-23-09565-f006:**
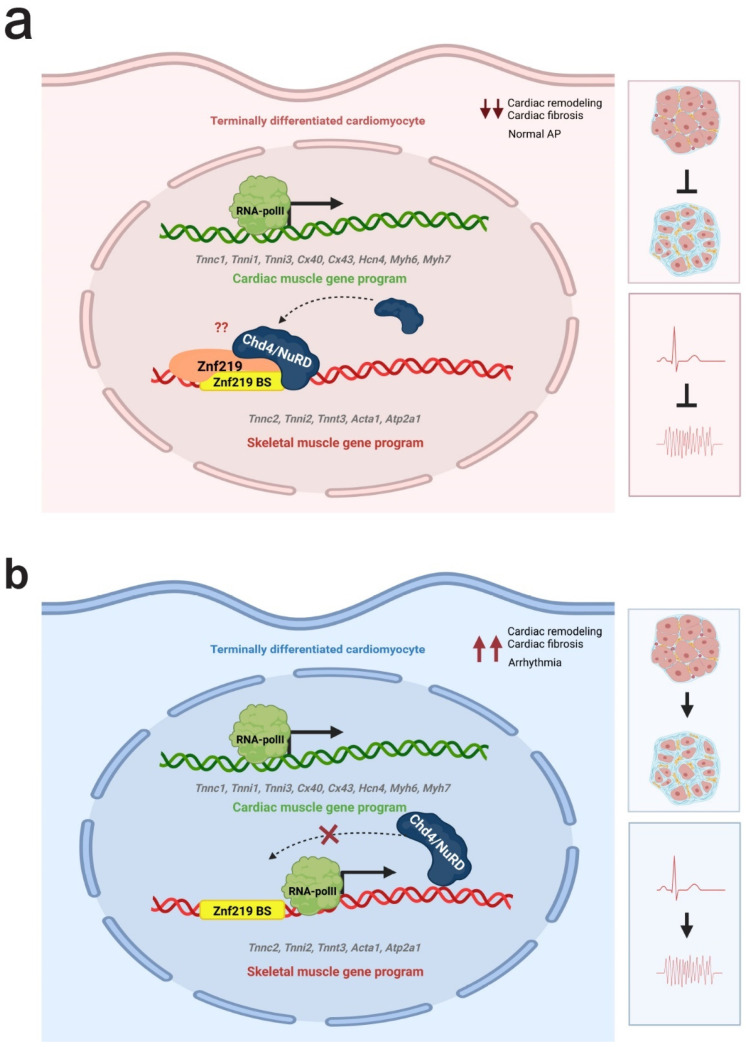
Model of the function of Znf219 in the mammalian heart. Proposed model of Znf219-Chd4/NuRD interplay in cardiac regulation. (**a**) Znf219/Chd4 complexes silence the expression of the skeletal-muscle gene program in cardiac tissue. (**b**) In the absence of Znf219 or Chd4, the skeletal-muscle gene program is activated ectopically in the heart, leading to fibrosis, conformational arrhythmias, and cardiac remodeling. This figure was created with biorender.com.

**Table 1 ijms-23-09565-t001:** Oligonucleotide primers used for qPCR.

Gene Name	Oligo Forward (Fw)	Oligo Reverse (Rv)
*m36B4 (Rplp0)*	GCGACCTGGAAGTCCAACTA	ATCTGCTGCATCTGCTTGG
*Znf219*	GGGACCCAGGCGAGATCC	CACGGCCCGCTTCAGATC
*Chd4*	CTGCTCAGCGGGAGTGAGG	GGGTCTCGAGGTTTCTTAGGCT
*Tnni2*	CACCTGAAGAGTGTGATGCTC	GGGCAGTGTTCTGACAGGTAG
*Acta1*	GTGACCACAGCTGAACGTG	CCAGGGAGGAGGAAGAGG
*Tnnc2*	GAGATGATCGCTGAGTTCAAG	GTCTGCCCTAGCATCCTCAT
*Atp2a1*	TTTGGCAGGAACGGAATG	AGCCTTGATCCTTTGCACTG
*Tnnt3*	ACAGATTGGCGGAGGAGAAG	CATGGAGGACAGAGCCTTTT
*Cx40*	GGAGGAAAGGAAGCAGAAGG	GACTGTGGAGTGCTTGTGGA
*Cacna1h*	ATTTCTGGCCGAACAGTCC	AGCATCTTGGAGGCCTTTG
*Hcn4*	TGACTTCAGATTTTACTGGGA	TTGAAGACGATCCAGGGTGT
*Kcnh2*	TCGCTTTCTCAGGTTTCCCA	GCCTGGATCTGAGCCATGT
*Scn5a*	TCGTCATGGCATACACAACT	GTCTTCAGGCCTGAAATGA
*Scn10a*	TTGACAACTTCAATCAACAGAA	TGGTACTTATTCAAAGGCCGT
*Col1a1*	GCTCCTCTTAGGGGCCACT	CCACGTCTCACCATTGGGG
*Col3a1*	CTGTAACATGGAAACTGGGGAAA	CCATAGCTGAACTGAAAACCACC

**Table 2 ijms-23-09565-t002:** Primary and secondary antibodies used and their respective dilutions.

Antibody	Host	Dilution	Secondary	Dilution
α-Flag	Mouse	1:500	α-mouse	1:5000 or 1:3000
α-Myc	Mouse	1:15	α-mouse	1:5000
α-tubulina	Mouse	1:400,000	α-mouse	1:5000
α-CHD4	Rabbit	1:500	α-rabbit	1:5000
α-Znf219	Mouse	1:500	α-mouse	1:2000
α-Tnni2	Rabbit	1:500	α-rabbit	1:3000
α-Tnnt3	Mouse	1:500	α-mouse	1:3000

## Data Availability

All the data supporting this study are available within the article and its Appendix A or can be obtained from the corresponding author upon reasonable request.

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
