# Peer review of "Interplay between the Chd4/NuRD Complex and the Transcription Factor Znf219 Controls Cardiac Cell Identity"

_ijms, 2022, doi:10.3390/ijms23179565_

Round 1
Reviewer 1 Report
The article under review represents a study investigating the association between the chromatin remodeler complex Chd4/NuRD and the transcription factor Znf219. The authors conclude that there is a strong interaction between Znf219 and Chd4 having an impact on mammalian cardiomyocyte identity and myocardial contraction. While the rationale behind conducting the study is valid, the following aspects should be addressed:
· In the abstract and in the introduction, please clarify what the aims of the current study were.
· In the methods section, please state how many mice were used in the different experiments and if there were any unexpected deaths.
· In the statistics section, the authors write: “The significance of differences between WT and corresponding mutant groups was assessed by two-tail Student’s t-test, one-way or two-way ANOVA, as stated in each figure legend.” It appears that only parametric statistical tests have been used, which assume a normal distribution of data. Has the distribution of the data been checked?
· Please report what statistics software and version were used to conduct the analyses and generate figures.
· In the results section, there are multiple concluding statements about the data, such as: “Taken together, these data strongly suggest that Znf219, like Chd4/NuRD, control de repression of the skeletal muscle gene program in the heart to thus maintain cardiac identity.” Such statements might be more appropriate to include in the discussion section.
· Several recently published studies (such as PMID: 34095631) have shown that machine learning methods could have additional benefits when analyzing cardiovascular transcriptomics data compared to only using traditional statistics models. It would be beneficial to mention this in the discussion section as a future perspective.
· Please add a paragraph on study strengths and limitations.
Author Response
Dear Reviewer,
I would first like to thank the reviewer for a thorough revision of the MS. These changes would greatly improve the MS. I would also like to point out that our in-house expert, who is an English-Speaking editor, has checked the grammar of the text and made the pertinent corrections. For easiness, these extensive changes are not marked in the text, but the reviewer will notice the improvement of the English all over the text. I have then numbered the questions of the reviewer and answered them (Ans:) one by one, marking in the text the changes in yellow and with the Word tracking system. For comprehension, we will be pointing the places in the MS where we reflect the changes made.
Comments and Suggestions for Authors
The article under review represents a study investigating the association between the chromatin remodeler complex Chd4/NuRD and the transcription factor Znf219. The authors conclude that there is a strong interaction between Znf219 and Chd4 having an impact on mammalian cardiomyocyte identity and myocardial contraction. While the rationale behind conducting the study is valid, the following aspects should be addressed:
1) In the abstract and in the introduction, please clarify what the aims of the current study were.
Ans.: Thank you for the suggestion. We have now included the aims of the study in both sections. Lane 26 of the abstract and 105 of the introduction.
2) In the methods section, please state how many mice were used in the different experiments and if there were any unexpected deaths.
Ans: Thank you for this remark. We agree it is important to add the animals involved in each experiment. We now mention the number of animals in the text and as follows:
For the protein and relative mRNA and protein quantification of Znf219/Znf219 respectively, 4 animals were used per organ (n=4) (Fig2 a y b).
For embryo relative mRNA was quantified from 3 different pools, each pool was originated from 2-3 hearts (n=3)/ total hearts (n=64) (Fig 2e).
For co-IP of Chd4 and Znf219, pools of 3 WT and 3 Chd4Corin were used (Fig 1e).
A total of 22 animals were used for histology (Picrosirius, GFP and Hematoxylin-eosin and fluorescence scope), distributed in two experiments of 9 animals (4 ShScramble/5 ShZnf219) and 13 animals (6 ShScramble/7 ShZnf219). representative images of each group were added in Figure 3 and figure S5.
For hypertrophy analysis we used 10 ShScramble and 10 ShZnf219 for the analysis (n=20) (Fig 3c).
For Znf219 (n=10; ShScramble=5, ShZnf219 =5) and Chd4 knockdown analysis in mice (Fig 4a). To assess knockdown of Znf219 protein, as well as the upregulation of skeletal troponins (Fig 4d) we used ShScramble=3, ShZnf219 =3 (n=6) (Fig 4b). To assess the upregulation of Atp2a1 in the atrium, we used ShScramble=3, ShZnf219 =3 (n=6) (Fig4 e). To assess the expression of cardiac troponins we used 3 shScr and 6 shZnf219 (Fig 4c).
For fig 5 a b y c à ShScramble (n=6), ShZnf219 (n=9).
No unexpected deaths were observed during this study.
3) In the statistics section, the authors write: “The significance of differences between WT and corresponding mutant groups was assessed by two-tail Student’s t-test, one-way or two-way ANOVA, as stated in each figure legend.” It appears that only parametric statistical tests have been used, which assume a normal distribution of data. Has the distribution of the data been checked?
Ans: Thanks also for this concern that we were not aware of. Now we have subjected all the data to the normality or log-normality tests using GraphPad 8.0 software and QQ plots. All data passed the Shapiro-Wilk and Kolmogorov-Smirnov normality tests. We have added this paragraph to the M&M section (4.11) page 18, and the plots could be added if requested by the reviewer.
4) Please report what statistics software and version were used to conduct the analyses and generate figures.
Ans: GraphPad Prism 8.0 for quantitative data analysis; Adobe Illustrator CC 2018 to generate the figures and MetaMorph offline software for fibrosis quantification. We have also added this in the 4.11 section page 18.
5) In the results section, there are multiple concluding statements about the data, such as: “Taken together, these data strongly suggest that Znf219, like Chd4/NuRD, control de repression of the skeletal muscle gene program in the heart to thus maintain cardiac identity.” Such statements might be more appropriate to include in the discussion section.
Ans.: Thank you for remarking this. We have now removed most of the statements from results and moved them to the discussion (page 14, lanes 419-422).
6) Several recently published studies (such as PMID: 34095631) have shown that machine learning methods could have additional benefits when analyzing cardiovascular transcriptomics data compared to only using traditional statistics models. It would be beneficial to mention this in the discussion section as a future perspective.
Ans.: We would like to thank the reviewer for this comment. It is true that Artificial Intelligence (AI) is becoming increasingly used to analyze cardiovascular transcriptomics. We, in fact, used in the past “random forest” to build a predictor of atherosclerosis analyzing with three biomarkers. The model performed better than conventional logistic regression but it was applied to a population of 3,000 individuals, where building a predictor that works as a biomarker makes sense.
The paper cited by the reviewer is totally different: it applies AI to a very small population (36 individuals) and the AI avoids the problem of overfitting with conventional techniques (too many parameters for so few individuals), but it probably will not fit with our data. We discuss the putative use of these techniques in the future and included this in the discussion (Section 3.1 page 15 lanes 444-448 - see also answer to next question).
7) Please add a paragraph on study strengths and limitations.
Ans.: This is a very appropriate comment, since the study opened further avenues of research. In fact, we are performing some experiments to deepen in the study of the synergy of Ch4/NuRD, Znf219 and other transcription factors in heart homeostasis. We therefore have included a paragraph in the discussion section at the end of the MS stating both the strengths and the limitations of our study. Pages 14 and 15
It would be as follows:
3.1 Study strengths and limitations.
The lack of Znf219 cKO mice makes very difficult to study the role of this transcription factor in vivo. Our approach to the problem, infecting with adeno-associated viruses to knockdown the expression of this gene, has allowed us to corroborate that Znf219 is important in heart homeostasis, maintaining skeletal muscle genes silenced. The limitations of the study, however, can be found in the translation of our results to human cardiac pathophysiology, since our investigation is basic in nature. Chd4 mutations have been found in humans that translate into cardiac disorders, but more research would be needed in order to find mutations or miss-expression of Znf219 in humans. Another limitation of our study is that, although we have found a direct interaction between Znf219 and Chd4, we do not directly demonstrate in this work that there is a transcriptional cooperation between both proteins. We are currently performing Znf219 silencing experiments in Chd4 cKO mice to corroborate this hypothesis.
Our study highlights the power of reanalyzing existing transcriptomics datasets using appropriate bioinformatic tools. For the discovery of novel factors, we envision that this approach will be boosted in the near future with the use of more advanced statitical tools optimized to extract relevant information depending on the characteristics of the transcriptomics datasets, such as artifcial intelligence models.

Reviewer 2 Report
The authors identify a role for Znf219 in regulation of cardiac gene expression. Overall the study is interesting but could use some additional experimentation to improve rigor and bolster conclusions made in the manuscript. Specific comments are outlined below.
Major:
IgG controls are needed for immunoprecipitation experiments in Fig 1e. Why is there so much Chd4 expression in the Chd4 cKO heart (input)?
More rigorous analyses and quantification of cardiac fibrosis is needed to make any claims about the effect of Znf219 knockdown. Strange to conclude that 15% of shZnf219 mice present with interstitial fibrosis. All hearts have ECM and deposition of collagens and other components that increases in disease to a point that is detectable by histological methods but this is not a binary response (e.g. there is some interstitial fibrosis in the shScramble heart too). % fibrotic area needs to be quantified in several animals in order compare genotypic effects.
More rigorous analyses of cardiac hypertrophy in response to Znf219 knockdown is needed. There is a modest but statistically significant increase in HW/TL with Znf219 knockdown but cardiomyocyte surface area should be quantified in these hearts. Does knockdown of Znf219 in HL-1 cells cause them to hypertrophy in vivo?
Immunohistochemistry for Atp2a1 in Fig 4 is strange. Why is Atp2a1 only induced in such a small percentage of shZnf219 of cardiomyocytes (<5%)? Are these the cardiomyocytes transduced with the shRNA (GFP positive)? If so, this transduction efficiency is far lower than presented in Fig 3. Atp2a1 should give an SR/ER localization pattern.
Is there a concomitant downregulation of cardiac genes with the upregulation of skeletal muscle genes in the Znf219 knockdown hearts?
Minor:
Is Znf219 expressed in the atria or AV node? What is the proposed mechanism for delayed atrioventricular conduction time in shZnf219 mice?
Premature ventricular contractions observed in shZnf219 hearts (Fig 5a) are actually very common in wildtype mice.
The effects of Znf219 knockdown on cardiac hypertrophy and fibrosis in vivo may be difficult to determine given the mosaicism and ~50% transduction efficiency of the AAV9 shRNA (Fig 3d).
Knockdown of Znf219 in HL-1 cells (Fig 2f) is modest at best.
The choice of the Cre line (Corin-Cre) to investigate cardiac transcriptional regulation is strange given the extracardiac expression of corin, particularly given the availability of Chd4 conditional knockout mice with the Nkx-Cre.
What are the control genotypes for Chd4 cKO experiments? This should be clear in the methods and/or results section. Wildtype mice are not the appropriate control genotype.
Abbreviation of Connexin genes in the summary Fig is incorrect. Gene names are Gja (gap junction) or Cx40/Cx43 commonly used in the literature.
shRNA sequences need to be included in the methods.
HL-1 cells are not cardiomyocytes. Expression of cardiomyocyte markers shows that they have a cardiomyocyte-like phenotype but does not confirm that they “are indeed cardiomyocytes”.
Author Response
Dear reviewer,
I would first like to thank the reviewer to a thorough revision of the MS. These changes would greatly improve the MS. I would also like to point out that our in-house expert, who is an English-Speaking editor, has checked the grammar of the text and made the pertinent corrections. For easiness, these extensive changes are not marked in the text, but the reviewer will notice the improvement of the English all over the text. I have then numbered the questions of the reviewer and answered them (Ans:) one by one, marking in the text the changes in yellow and with the Word tracking system.
Comments and Suggestions for Authors
The authors identify a role for Znf219 in regulation of cardiac gene expression. Overall the study is interesting but could use some additional experimentation to improve rigor and bolster conclusions made in the manuscript. Specific comments are outlined below.
Major:
1) IgG controls are needed for immunoprecipitation experiments in Fig 1e. Why is there so much Chd4 expression in the Chd4 cKO heart (input)?
Ans.: We thank the comment of the reviewer, but I would like to convince him/her why it is better to use as a control Chd4 cKO heart than IgG.
Chd4 is actually expressed in all the heart cell types (see Gómez del Arco et al. 2016 Cell Metabolism). Chd4/corin cre mutant mice would only delete Chd4 in the fully-differentiated cardiomyocytes, but other cell types (fibroblasts, epicardium, smooth muscle cells, endocardium, etc.), would still express Chd4 in these mutants. Our intention was to discriminate between the proteins interacting with the NuRD complex in cardiomyocytes versus the other cell types. For that reason, we IPd Chd4 both in WT (Chd4F/F;corinwt/wt) versus Chd4 cKO (Chd4F/F;corincre/cre). Chd4 mutant hearts still presents expression of Chd4 from other non-cardiomyocytic cell types, with a NuRD complex still intact, as demonstrated in our IP, where presenting a concomitant reduction in peptides of the components of the NuRD complex, including those of Chd4. However, we find no presence of Znf219, or other proteins, in mutant hearts, so we assumed that this TF would only bind to the complex in cardiomyocytes. Moreover, the proteomic facility at CNIC (our collaborators), as experts in the field, recommended us to use the cKOs as control.
Additionally, in our hands, it is very difficult to select a good IgG control, since we have sometimes observed unspecific binding of some proteins of the NuRD complex with certain commercially available isotype controls. This is also the case with some sepharose or agarose beads.
2) More rigorous analyses and quantification of cardiac fibrosis is needed to make any claims about the effect of Znf219 knockdown. Strange to conclude that 15% of shZnf219 mice present with interstitial fibrosis. All hearts have ECM and deposition of collagens and other components that increases in disease to a point that is detectable by histological methods but this is not a binary response (e.g. there is some interstitial fibrosis in the shScramble heart too). % fibrotic area needs to be quantified in several animals in order compare genotypic effects.
Ans.: We thank the reviewer for this major concern. We agree with him/her that this was a major flaw of our study, because we did not properly quantify the fibrosis and it represented, indeed, very preliminary data. We have now performed SiriusRed in more animals and found that, actually, not 15%, but 30% (4 out of 13 animals) of them presented a clear (to the eye) fibrotic interstitial accumulation, although we have to say that the contingency analysis is still not significant (see new figure S5). We have also quantified the fibrotic area of 10 shScramble and 13 Znf219 KD animals by MetaMorph and found that there is a clear tendency (still not significant) to present more fibrosis in the hearts infected with AAV expressing shRNA for Znf219 (new Figure S5b; p=0.07).
I believe this might be due to the fact that we have analyzed the fibrosis at early stages of the kockdown, with not enough time to develop measurable fibrosis in all of the Znf219 knockdown mice. For this reason, we wanted to check earlier fibrotic indicators, as Collagens 1a and III, at the transcriptional level. In these experiments we find a tendency of increase in expression of Col1a (p=0.08), but a significant increase of colIII mRNA (New Figures S5d; p<0.04). We also include the sequences of the primers used in the qPCR in table 1.
Taken together, these data demonstrate that fibrosis is one of the features present in the hearts KD for Znf219. As mentioned, we now present these data in a Supplementary figure (S5), in which we also indicate the number of mice presenting fibrosis and pictures or the Sirius Red of 4 Sh-Scramble infected hearts and of the 4 sh-Znf219 infected ones, showing fibrosis (See results text at the end of page 7).
3) More rigorous analyses of cardiac hypertrophy in response to Znf219 knockdown is needed. There is a modest but statistically significant increase in HW/TL with Znf219 knockdown but cardiomyocyte surface area should be quantified in these hearts. Does knockdown of Znf219 in HL-1 cells cause them to hypertrophy in vivo?
Ans: This is a very important point raised by the reviewer. We agree that the hypertrophy is one of the features, together with fibrosis, that may indicate cardiac pathology. Unfortunately, this is actually a difficult task and the easier way is to measure of HW/TL ratio showing, as pointed out by the reviewer, a modest increase in the weight of the hearts with a knockdown Znf219. We did try to perform Wheat Germ Agglutinin (WGA) staining to measure cardiomyocyte area but, in our hands, was very difficult to conclude anything, since it is almost impossible to ascribe the measure of the cardiomyocyte area stained with WGA, with the infection in that particular cardiomyocyte, and also considering that that particular cardiomyocyte may have or not a complete knockdown of Znf219, because the infection levels also varies. With these tests, we were unable to reach any conclusive results. Moreover, as in the case of fibrosis, at 8 weeks of age, after a short and mosaic Znf219 KD, probably hypertrophy is not the best way to measure cardiac pathology. We would probably need to increase the percentage of infection, and the age of the mice.
The reviewer also indicates that HL-1 could be also suitable to measure hypertrophy but, again, this is also a very difficult task because these cells have to be maintained with a culture media supplemented with hypertrophic factors, such as norepinephrine.
We apologize for the lack of data in this particular issue.
4) Immunohistochemistry for Atp2a1 in Fig 4 is strange. Why is Atp2a1 only induced in such a small percentage of shZnf219 of cardiomyocytes (<5%)? Are these the cardiomyocytes transduced with the shRNA (GFP positive)? If so, this transduction efficiency is far lower than presented in Fig 3. Atp2a1 should give an SR/ER localization pattern.
Ans.: Thanks again for the comment. We actually had at first the same concern. In our previous study (Gómez del Arco et al. 2016 Cell Met.- (Figure 3F) of that report), we found Serca1 (Atp2a1) protein ectopically expressed majorly in the auricles of Chd4 cKO (deleted with a-MHC-cre or corin-cre). Even though these mutant mice had a deleted Chd4 alike in auricular and ventricular cardiomyocytes, we did see only a few ventricular cardiomyocytes (mostly of the right ventricle) expressing Serca1. We could not explain this in our previous report, so we cannot explain it in the present study, although the pattern of expression of Serca1 is, like in the Chd4 cKO, mostly found in the auricles. In fact, by mRNA, we do not find upregulation of Atp2a1 in the Znf219 KD hearts (we mention this in the results).
Regarding the cellular localization of Serca1, we do not believe IHC is the right technique to ascertain this and a cytoplasmic staining is observed, like in our previous report (Gómez del Arco, 2016 Cell Met.).
We might be able to answer this question in our next work, since we are now Knocking down Znf219 in a Chd4 background. Our hypothesis is that Chd4 and Znf219 may synergize to regulate Atp2a1 transcription.
5) Is there a concomitant downregulation of cardiac genes with the upregulation of skeletal muscle genes in the Znf219 knockdown hearts?
Ans.: This is a very pertinent question, which we forgot to reflect in the MS. The simple answer is no. There is no change in the sarcomeric cardiac genes, when Znf219 is KD in the heart, as we also observed when we deleted Chd4 in our previous report (Gómez del Arco, 2016. Cell Met.).
We usually perform qPCR controls of cardiac tissue (Tnnc1 and Tnnt2) and never found any change. We have now performed additional cardiac genes (Actc1 and Tnni3) and found no change either. We present these data in a supplementary figure (S6b) and result text page 10, last paragraph of section 2.4.
We also show here the plots.
Minor:
1) Is Znf219 expressed in the atria or AV node? What is the proposed mechanism for delayed atrioventricular conduction time in shZnf219 mice?
Ans.: This a very god point of the reviewer that we actually would like to know too. Although we have not checked a specific cardiac localization for Znf219, we have checked the expression of some AVN markers when Znf219 is knockdown (i.e Hcn4) with no change (Figure S6a). We have also checked the expression of some Ion channels, but found no changes either (Figure S6a). We, therefore, explain the delay in the propagation of the electrical impulse and contraction, in the aberrant expression of skeletal-muscle genes.
It is not possible to check at this time the expression of Znf219 in the AVN by IHC, since there are not commercially available antibodies. We could not discriminate the levels of expression of Znf219 in the AVN, the SAN, or the Purkinje fibers with the infected AAV9 shRNA-Znf219. We, therefore, cannot satisfactorily answer this question.
2) Premature ventricular contractions observed in shZnf219 hearts (Fig 5a) are actually very common in wildtype mice.
Ans.: It is true that some WT animals of certain strains present premature ventricular contractions, but we have not observed this in our WT animals. We actually have performed most of the experiments in the CD1 background
3) The effects of Znf219 knockdown on cardiac hypertrophy and fibrosis in vivo may be difficult to determine given the mosaicism and ~50% transduction efficiency of the AAV9 shRNA (Fig 3d).
Ans: We agree with the reviewer on this point. We could need to increase the infection efficiency and/or the time of analysis. We believe 8 weeks of Znf219 KD is not enough to measure fibrosis. Even though, we have tried and there are indications that support our statement (see answer to major point 2).
4) Knockdown of Znf219 in HL-1 cells (Fig 2f) is modest at best.
Ans: We agree with the reviewer that the KD with the shRNA for Znf219 is modest, as we pointed out in the MS. We, in fact, discuss that this might be the reason why we see a modest but significant upregulation of the skeletal muscle genes in HL-1 cells KD for Znf219. Before deciding the amount of viral particles used in this work, we performed a series of experiments in which we progressively increased the amount of viral particles of three different commercially available lentiviral shRNAs. All of them partially KD Znf219 transcription, but the one used in this study was the best, as it was the virus concentration used. Using more concentrated viruses had deleterious consequences for the cells. We speculate that Znf219 half-life might be very long in these cells, or that the sh-RNA is not perfect to completely knockdown Znf219.
5) The choice of the Cre line (Corin-Cre) to investigate cardiac transcriptional regulation is strange given the extracardiac expression of corin, particularly given the availability of Chd4 conditional knockout mice with the Nkx-Cre.
Ans: The reviewer is right considering that the corin-cre is not the perfect choice to study the role of Chd4 cKO, but our intention in this study was using terminally differentiated cardiomyocyte of Chd4 cKO hearts, just to Immunoprecipitating NuRD complex. In our previous report (Gómez del Arco et al, Cell Met. 2016), we extensively studied Chd4 role in cardiac development and homeostasis, deleting Chd4 with Nkx2-5 cre, c-TnT-cre, a-MHC-cre and corin-cre. We could not use Nkx2-5 cre in the present study because the mutant animals die during embryogenesis (ED14.5) and, again, our intention was to study Chd4/NuRD interactors in terminally differentiated cardiomyocytes.
6) What are the control genotypes for Chd4 cKO experiments? This should be clear in the methods and/or results section. Wildtype mice are not the appropriate control genotype.
Ans: I would like to thank the reviewer to notice that we have not mention in the text the mice used as WT in the Chd4 cKO IP experiments. These mice are littermates of the mutant mice, with Chd4 floxed and no cre. We actually mentioned it in the M&M, in the section 4.4 (lane 479), but we have now also included it in the results (lanes124 and142) and legend of Figure 1 (lane 160).
I apologize for this mistake.
7) Abbreviation of Connexin genes in the summary Fig is incorrect. Gene names are Gja (gap junction) or Cx40/Cx43 commonly used in the literature.
Ans: Thanks again to the reviewer for noticing this mistake and we apologize. We have now corrected in the text and in the Figure 6.
8) shRNA sequences need to be included in the methods.
Ans: We include the sequences of lentiviral and Adeno-Associated sh-RNAs in a new Table S4, so we mention it in several places in the results (lanes 185 and 225) and M&M (section 4.2, lane 472 v)
9) HL-1 cells are not cardiomyocytes. Expression of cardiomyocyte markers shows that they have a cardiomyocyte-like phenotype but does not confirm that they “are indeed cardiomyocytes”.
Ans: We agree with the reviewer that HL-1 cells are not primary cardiomyocytes and apologize to infer this in the manuscript. These cells are considered a “cardiomyocytic cell line” or a so-called “cardiac muscle cell line” by Dr. Claycomb. They are derived from mouse atrial cardiomyocytes. It is true that this is not the perfect model, but it is widely used due to the lack of cardiac cell models of terminally differentiated cardiomyocytes, which are also difficult to obtain when derived from iPSCs. They express cardiac sarcomeric proteins and they beat. We have changed HL-1 cardiomyocyte to HL-1 cells only once in the text (lane 368 of the discussion) and we made sure to called them just HL-1 cells or cardiomyocytic cell line throughout the text.

Round 2
Reviewer 2 Report
The manuscript is improved but several to most of the prior concerns have not been sufficiently addressed.
1. IgG controls are needed for immunoprecipitation experiments in Fig 1e. Although Chd4 cKO are a good control, it does not control for nonspecific immunopreciptiation of Znf219 by nonspecific sticking to agarose/sepharose/antibody and thus IgG controls are also needed. In fact, the rationale provided by the authors (prior observation of NuRD complex proteins to IgG, sapharos/agarose) even further substantiate the need for IgG controls and suggest the detection of Znf219 in the Chd4 fl/fl immunoprecipitate may very well be nonspecific.
2. Rigorous analyses of cardiac hypertrophy and cardiomyocyte cell surface area are needed. WGA staining of cardiac sections to quantify CM surface area is very simple and done in. 10s of 1000s of published studies. Costaining for GFP can be performed (as in Fig 3d) to assess CM area only in cells infected with shZnf219, which should give sensitive results and enable definitive determination of whether knockdown of Znf219 induces cardiomyocyte hypertrophy in vivo.
3. It appears from revision experiments that there was only a trend towards increased fibrosis in shZnf219 hearts so conclusions need to be toned down accordingly. Moreover, data needs to be presented better. % fibrotic area from Picrosirius Stained red should be quantified. Y-axis label on Fig S5b is "staining area relative to expression". What exactly is this staining area normalized to? Data should be quantified as % fibrosis (PSR positive area/ total myocardial area evaluated). Data in Fig S5c should be removed. As discussed in the previous reviewer report, cardiac fibrosis is not a binary response. Label for quantification of Collagen 3 transcripts in Fig S5d needs to be appropriately labeled with the correct gene names. Is this Col1a3? What about the Col1a gene on the left- was this with primers specific for Col1a1- there are several Col1a isoforms.
5. Assessment of cardiomyocyte hypertrophy in vitro with acute knockdown of shZnf219 would be quite helpful in determining if Znf219 truly has antihypertrophic activity. Many studies assess hypertrophy in HL-1 cells, but as described previously, this is not a great CM model anyways but could alternatively be easily performed in rat neonatal cardiomyocytes.
